# Emerging Therapies for Charcot-Marie-Tooth Inherited Neuropathies

**DOI:** 10.3390/ijms22116048

**Published:** 2021-06-03

**Authors:** Marina Stavrou, Irene Sargiannidou, Elena Georgiou, Alexia Kagiava, Kleopas A. Kleopa

**Affiliations:** 1Neuroscience Department, The Cyprus Institute of Neurology and Genetics, Cyprus School of Molecular Medicine, Nicosia 2371, Cyprus; stavroum@cing.ac.cy (M.S.); irenes@cing.ac.cy (I.S.); elenag@cing.ac.cy (E.G.); alexiak@cing.ac.cy (A.K.); 2Center for Neuromuscular Diseases, The Cyprus Institute of Neurology and Genetics, Cyprus School of Molecular Medicine, Nicosia 2371, Cyprus

**Keywords:** Charcot-Marie-Tooth disease, inherited neuropathy, gene therapy, axonal degeneration, biomarkers

## Abstract

Inherited neuropathies known as Charcot-Marie-Tooth (CMT) disease are genetically heterogeneous disorders affecting the peripheral nerves, causing significant and slowly progressive disability over the lifespan. The discovery of their diverse molecular genetic mechanisms over the past three decades has provided the basis for developing a wide range of therapeutics, leading to an exciting era of finding treatments for this, until now, incurable group of diseases. Many treatment approaches, including gene silencing and gene replacement therapies, as well as small molecule treatments are currently in preclinical testing while several have also reached clinical trial stage. Some of the treatment approaches are disease-specific targeted to the unique disease mechanism of each CMT form, while other therapeutics target common pathways shared by several or all CMT types. As promising treatments reach the stage of clinical translation, optimal outcome measures, novel biomarkers and appropriate trial designs are crucial in order to facilitate successful testing and validation of novel treatments for CMT patients.

## 1. Clinical Features, Classification and Genetics of CMT Neuropathies

Charcot-Marie-Tooth disease (CMT) is an eponym for a large and genetically highly heterogeneous group of inherited peripheral neuropathies, recognized since the 1800s. Collectively, CMT neuropathies have a prevalence of 1 in 2500 persons, and are therefore one of the commonest type of neurogenetic diseases world-wide [1,2]. As with many long recognized chronic neurogenetic and neuromuscular disorders, the treatment of CMT neuropathies has been supportive and symptomatic for over 100 years. However, based on the recent discovery of the underlying molecular-genetic causes of the different disease types, new therapies have begun to emerge that are currently at different stages of development, both preclinical and clinical. This review summarizes recent developments in both preclinical as well as clinical stage therapies for different types of CMT neuropathies, along with some aspects of clinical trial readiness that will enable translation of experimental therapeutics into clinical practice.

CMT neuropathies usually present in childhood or adolescence, however there is a wide range of onset from infancy to late adulthood. In most CMT types, progression is very slow and disability accumulates gradually over the lifespan without being life-threatening. Affected individuals typically have distal muscle weakness and atrophy, weak ankle dorsiflexion, depressed tendon reflexes, and high-arched feet, also known as *pes cavus* deformity. Mild to moderate distal sensory loss, typically in a symmetric, so-called stocking-glove distribution, usually accompanies the muscle weakness [3]. In most cases these symptoms are painless, but for some patients CMT neuropathy can be painful. CMT neuropathies are generally considered non-syndromic. However, additional clinical manifestations such as hearing loss, optic neuropathy, or central nervous system dysfunction may occur in certain CMT types [4].

CMT neuropathies have been classified clinically and electrophysiologically in the pre-molecular era using at least three different criteria: First, according to severity and age of onset; second, whether they are primarily demyelinating or axonal; and third, whether they affect both motor and sensory, or almost exclusively one of the two fiber types [4,5,6,7,8]. The classification into demyelinating or axonal CMT types is based on the findings of nerve conduction studies. The demyelinating types (CMT1 and CMT4) are characterized by motor nerve conduction velocities (MNCV) in the upper limb below 35 m/s. In contrast, the axonal (non-demyelinating) CMT2 or CMT3 types are defined by MNCV above 45 m/s. A third category, the dominant intermediate CMT (DI-CMT), had to be introduced because there is a group of CMT types that is clinically indistinguishable from CMT1 or CMT2 as it shows MNCVs between 35–45 m/s and thus cannot be classified with certainty as demyelinating or axonal. Intermediate slowing of MNCV is also characteristic of the commonest X-linked CMT type, CMT1X. Variability in MNCV can be found between affected individuals in the same family with intermediate CMT.

The most severe inherited neuropathies manifest at birth, also known as congenital hypomyelinating neuropathy (CHN), with hypotonia and arthrogryposis caused by reduced mobility in utero, swallowing or respiratory difficulties, and an overall poor prognosis. Due to the severe effects of the CHN mutations in affected patients, myelination is not achieved and nerve conduction studies remain below 5 m/s. When the demyelinating neuropathy manifests in infancy with delayed motor milestones before 3 years of age, the term Déjérine-Sottas neuropathy (DSN) or syndrome (DSS) is used. There is a continuous spectrum of clinical phenotypes overlapping with childhood onset and relatively progressive course that is typical of the recessive demyelinating (CMT4) or axonal (CMT3) neuropathies [2,9].

CMT1 describes the most common and dominantly inherited demyelinating CMT types, while the term CMT2 describes a large group of dominantly inherited axonal neuropathies with variable severity. Even within the same CMT1 or CMT2 type, phenotypic variability is common and ranges from early childhood to late adulthood onset with different rates of progression. Affected individuals usually become symptomatic between ages 5 and 25 years, while fewer than 5% become wheelchair dependent and lifespan is not significantly shortened. All these clinical characteristics are particularly relevant for planning and testing novel treatments for CMT neuropathies, since outcome measures are influenced by the natural history and the inhomogeneous degree of severity within each genetically defined CMT type. Disease modifying genes have been discussed in this context, especially in CMT1A patients [10]. A more detailed clinical classification and phenotypes of CMT neuropathies are presented in other reviews [4,5]. In addition to the CMT disorders affecting both motor and sensory function, there are additional categories that include purely motor neuropathies, named distal hereditary motor neuropathies (dHMN) [11], as well as purely sensory (HSN) or sensory and autonomic (HSAN) neuropathies [12,13].

Over the last decades, additional criteria for classification of CMT neuropathies have been introduced based on the mutations in specific genes [14]. Clinical phenotypes are mostly explained by the molecular-genetic background, in that mutations in genes highly expressed in Schwann cells (SCs) typically cause demyelinating neuropathy, whereas mutations in genes that are highly expressed in neurons are predicted to cause axonal neuropathy. However, variable cellular effects of different mutations in the same gene may cause diverse phenotypes spanning the clinical categories. The molecular-genetic mechanisms of inherited neuropathies have been reviewed in detail in other publications [15,16,17,18] and are continuously updated in relevant databases such as the Online Mendelian Inheritance in Man (OMIM; http://www.ncbi.nlm.nih.gov/omim, last accessed: 27 April 2021), in the Gene Reviews (http://www.ncbi.nlm.nih.gov/books/NBK1358/, last accessed: 27 April 2021), and in the inherited neuropathy variant browser (http://hihg.med.miami.edu/code/http/cmt/public_html/index.html#/, last accessed: 27 April 2021).

Cellular mechanisms can be diverse in different CMT neuropathies and therefore understanding the molecular genetic basis of each CMT type is essential before developing any treatment approaches. Emerging therapies for CMT neuropathies can be classified into disease specific, which target the cause of the neuropathy, and to non-disease specific, which target downstream final common pathways, mostly mechanisms of axonal degeneration. Below we discuss currently studied therapies for the most common as well as for some uncommon CMT types, following a brief introduction to their molecular genetic basis that guides treatment approaches.

## 2. Overview of Therapeutic Approaches for CMT Neuropathies

### 2.1. Gene Therapy

Gene therapy refers to the delivery of genetic material to a subject mostly via viral vectors. A great spectrum of viral vectors [19] and vector capsid serotypes [20] has been developed to mediate gene delivery to different types of cells. The advantage of viral vector gene therapy is the potential to cross the blood-brain-barrier (BBB) as well as the blood-nerve-barrier (BNB), while also being a one-off treatment that provides long lasting effects. CMT types that result in either absent or nonfunctional proteins can be treated by adding the wild type (WT) version of the corresponding gene. In types caused by increased gene dosage effect, silencing the gene expression is the main goal, while in types with other toxic gain-of-function mechanisms gene editing or allele-specific silencing are considered.

Lentiviral vectors were one of the first vehicles to provide efficient delivery of therapeutic genes to treat CMT models [21,22,23,24,25,26]. Although clinical trials have been employed to test the local use of integrating lentiviral vectors for other diseases [27,28], their translatability for CMT neuropathies remains limited due to their integration into the host genome, making the direct systemic application problematic. The higher gene expression levels provided by adeno-associated viral (AAV) vectors and their episomal persistence in cells offers a more translatable method to overcome the risks of lentiviral incorporation into to the host genome [17]. Since both neurons as well as SCs targeted by CMT therapies are highly differentiated and not proliferating, the episomal persistence of AAVs without integration into the hosts’ genome will not affect the stability of the treatment. AAV vectors have been employed in promising preclinical research for treating CMT neuropathies either by expression of trophic factors or by targeting the responsible genes in neurons or SCs [29,30,31,32,33,34]. AAV1 is already used for a CMT1A clinical trial (NCT03520751), while other serotypes such as AAV9 have been clinically implemented for other neuromuscular diseases, most notably for spinal muscular atrophy (NCT03306277).

Silencing overexpressed or mutated genes is a promising approach to ameliorate the phenotype of many CMT forms. RNA-interference is a well-characterized method that includes small interfering RNA (siRNA), short hairpin RNA (shRNA), and microRNA (miRs). Those are small (usually ~19–22 nucleotides long) RNA structures that specifically bind on the transcript messenger RNA (mRNA) of interest in order to either degrade it or block its translation [35,36]. Antisense oligonucleotides (ASOs) are single-stranded synthetic nucleic acids that specifically bind on mRNA sequences and promote their RNAseH-depended degradation [37]. An alternative to ASOs, the antiparallel triplex-forming oligonucleotides, bind to the major groove of duplex DNA of interest [38]. Another approach for gene silencing is the use of clustered regularly interspaced short palindromic repeats (CRISPR) working in co-cooperation with Cas genes in order to disrupt, delete, add or replace a sequence of the targeted DNA or RNA regions [39,40].

A major challenge in translating gene therapy approaches for CMT neuropathies is the optimal and clinically translatable route of administration to achieve adequate biodistribution in the peripheral nervous system (PNS). For treating demyelinating neuropathies, transduction of as many and as widespread SCs as possible would be necessary. Local administration of therapeutic agents by direct intraneural injection is one possibility that has been explored using different vectors [21,26,41,42]. This approach is limited by the lack of biodistribution beyond the injected nerve and the need for multiple neurosurgical procedures. Intramuscular delivery has been mainly used to deliver vectors expressing trophic factors, such as NT-3 [29,43], while it does not provide adequate biodistribution to the nerves. Intravenous delivery of therapeutic agents may be an alternative method for clinical application providing access to peripheral nerves using vectors that can pass the BNB. However, high vector doses needed for this approach generate concerns about cost, immunogenicity and other unwanted effects, including liver toxicity [44,45,46]. Lumbar intrathecal injection of viral vectors expressing therapeutic genes is another clinically applicable delivery method that has been shown to provide an adequate biodistribution to the spinal roots and peripheral nerves, although a gradient can be observed towards distal nerves [24,25,34,47].

### 2.2. Small Molecule Therapies

Most drug therapy efforts for CMT neuropathies focus on the final common pathway of axonal degeneration, which occurs as a primary mechanism in axonal CMTs and as a secondary consequence in demyelinating CMT forms. In all cases, it correlates with disability and provides a meaningful therapeutic target [48]. As multiple pathways involved in axonal injury and degeneration are being discovered [49], druggable targets are identified. Several of them are in stages of preclinical or clinical testing for different CMT types, including CMT1 and CMT2. Axonal transport defects, altered mitochondrial dynamics, programmed axon degeneration, and impaired axon regeneration are common aspects of pathology in CMT neuropathies [50] and provide opportunities for drug based therapies that are discussed below for representative CMT forms.

## 3. Emerging Treatments for Demyelinating CMT Neuropathies

### 3.1. CMT1A

CMT1A is the commonest inherited demyelinating peripheral neuropathy, caused by *PMP22* gene duplication. The 1.4 Mb tandem intra-chromosomal duplication on chromosome 17p11.2-p12 results in three gene copies translated into PMP22 protein [51,52,53,54,55]. PMP22 is a 22-kDa intrinsic tetraspan glycoprotein primarily produced by myelinating SCs during development and makes 2–5% of PNS compact myelin [18,56]. This protein is crucial for SC growth and differentiation, myelogenesis, myelin thickness and maintenance of PNS axons and myelin [57,58,59]. PMP22 is also a regulator of cholesterol content in lipid rafts [60]. The increased amount of PMP22 protein creates a gene dosage effect that destabilizes the myelin sheath structure leading to demyelination and ultimately to secondary axonal loss and disability. These mechanisms have been demonstrated in rodent models of CMT1A including the transgenic rat, as well as Tr-J, C3, C22 and C61, JP18 and JP18/JY13 mice [61,62]. Hence, most efforts tested in experimental models to treat CMT1A have been focused on reducing the amount of PMP22 (Table 1). Some approaches have also been evaluated in clinical trials, however at present there is no cure for CMT1A but only symptomatic treatment (Table 1).

#### 3.1.1. Gene Therapy Approaches for CMT1A

Gene therapy approaches have been mostly designed to reduce *PMP22* overexpression at the DNA or mRNA level, while additional non-*PMP22* targeting methods have also been proposed.

*RNA-interference* (*RNAi*) has been one of the main targets in CMT1A gene therapies. Non-viral intraperitoneal delivery of synthetic siRNA showed that there was specific binding on the mutated *Pmp22*-L16P allele of Tr-J mouse that ameliorates neuropathic features [76]. Recently, intravenous injection of *PMP22*-targeting siRNA conjugated to squalenoyl nanoparticles in JP18 and JP18/JY13 mice modeling CMT1A decreased PMP22 levels and improved locomotor activity, electrophysiological parameters, myelination and neurofilament levels [44]. However, this treatment approach has short-lasting effects and requires repeated injections. Long lasting RNAi treatment was achieved by intraneural injection of an AAV2/9 vector expressing murine *Pmp22*-targeting shRNA to treat CMT1A rat pups [42]. This approach normalized MPZ and PMP22 protein levels and improved myelination and function. Intraneural injection allowed limited viral transduction in off-target organs and did not stimulate significant immune response. Whether this method would have the same beneficiary effects in older diseased animals remains unclear, along with the issue of limited clinical translatability of the intraneural injection.

Based on the knowledge that SCs express endogenous miRs that specifically bind on genes to regulate their expression [91,92,93,94,95,96,97,98], miRs with regulatory effects on *PMP22* have been tested for silencing. Intraneural administration into C22 mice of a lentiviral vector expressing miR-318 downregulated *PMP22* mRNA and reduced PMP22 protein levels resulting in improved behavioral, electrophysiological and histological parameters of this CMT1A model [26]. AAV2-miR-29a infection of C22 primary mouse SCs and human dermal fibroblast cultures also reduced *PMP22* transcript and restored mitotic activity [30].

*Antisense Oligonucleotides (ASOs)* provide an alternative tool to silence the *PMP22* transcript that requires repeated dosing. Subcutaneous administration of *PMP22*-targeting ASOs in C22 mice and in the CMT1A rat reduced the mRNA levels of both human and murine *Pmp22* and improved functional and morphological abnormalities of CMT1A rodent models in a dose-depended manner [78]. Antiparallel triplex-forming oligonucleotides were also designed to bind to *PMP22* promoters but no *PMP22* silencing was documented [79].

*CRISPR/Cas9* approaches aim at targeting regulatory elements in the *PMP22* gene to reduce transcription. CRISPR/Cas9 was used in a rat SC line in order to delete an upstream region of *PMP22* gene, which is predicted to host an enhancer or a promoter of the gene. Deletion of this region resulted in a reduction of *Pmp22* mRNA levels [37]. Likewise, CRISPR/Cas9-mediated deletion of the TATA-box promoter of *PMP22* gene in C22 mice using non-viral intraneural injections also downregulated *Pmp22* mRNA and improved nerve pathology [41]. Despite these promising studies, off target effects of gene editing approaches remain a concern and more flexible RNA-level editing methods may provide an emerging alternative [99].

*Indirect approaches bypassing the PMP22 gene dosage effect* to treat CMT1A include the downregulation of purinoceptor P2X7. P2X7 was found to have enhanced interaction with overproduced PMP22, causing increased extracellular Ca^2+^ influx into the SCs leading to functional derangement. siRNA and shRNA mediated downregulation of P2X7 in CMT1A rat SC co-cultures with dorsal root ganglion (DRG) neurons reduced Ca^2+^ influx, increased the expression of myelin-related proteins and improved SCs function, while P2X7 pharmacological inhibition had the same benefits [77].

*Neurotrophin-3 (NT-3)* is a neurotrophic factor crucial for SCs autocrine survival and regeneration [32,100]. The first attempt to employ NT-3 in CMT1A therapy was done by regular subcutaneous injections of NT-3 peptide in Tr-J and in immune-incompetent mice harboring xenografts of CMT1A patients, as well as in a group of 8 CMT1A patients [43]. For both animal models and patients, NT-3 exogenous administration resulted in augmented axonal regeneration, being in line with improved neuropathy score and sensory deficits. To overcome the necessity of repeated injections, *NTF3* cDNA encoding NT-3 was subsequently packaged into an AAV1 virus that was intramuscularly injected into the Tr-J model [29]. This resulted in improved myelination, motor function, histopathology and electrophysiological outcomes lasting for up to 40 weeks post-injection. Currently, scAAV1.tMCK.NTF3 is in phase I/IIa clinical trial (NCT03520751) employing bilateral intramuscular injections in CMT1A patients.

*VM202* (*Engensis*) refers to a non-viral vector which expresses a novel genomic cDNA hybrid of human hepatocyte growth factor (HGF) [74]. HGF was shown to promote peripheral nerve regeneration by stimulating SC repair [75]. VM202 was granted an orphan drug designation in 2014 and was categorized in fast track in 2016 by Food and Drug Administration (FDA). Clinical trials have already been conducted for this drug for other neurological conditions [101,102]. Although results from the CMT1A study have not been published yet, repeated intramuscular injections of the vector in ischemic heart disease [103] and amyotrophic lateral sclerosis [104] patients showed a decline of VM202 beneficial effects after a few months, indicating that this may be a short-lasting symptomatic treatment.

#### 3.1.2. Drug-Based Therapies for CMT1A

As in gene therapy approaches, drug therapies for CMT1A are intended to reduce the toxic effects of overexpressed *PMP22*. At their majority, drug therapies do not specifically target the expression of a gene and their targets may extend beyond current knowledge, so that long-term efficacy and safety remain to be addressed.

*PXT3003* is a liquid orally administrated cocktail of baclofen, naltrexone and sorbitol, mode-of-action and potency of which are not fully understood. PXT3003 was shown to decrease *Pmp22* mRNA levels, induce myelination, improve misbalanced PI3K-AKT/MEK-ERK signaling pathways, increase the number of functional neuromuscular junctions, promote Schwan cell differentiation and improve distal motor latencies and behavioral performance in the transgenic rat model of CMT1A [63,64,65]. However, these effects were not accompanied by major improvements in myelin thickness and internodal length [65]. A phase II clinical trial showed no significant improvement in neuropathy scores after a low dose treatment, but some improvement with the highest dose [105]. Although PXT3003 safety and tolerability were already confirmed in patients receiving the drug for one year [105] (NCT01401257), at the moment a clinical trial is assessing the long term safety and tolerability of the drug in CMT1A patients (NCT03023540). The phase III trial raised concerns about the stability of PXT3003 high dose which led to its termination (NCT02579759). A new clinical trial was launched in 2021 (NCT04762758) in which a fixed dose of PXT3003 is orally taken twice a day for 16 months. It is hoped that the outcome of this study will determine the potential of this treatment in CMT1A.

*Ascorbic acid* (Vitamin C) is known for its antioxidant, neuromodulator action and crucial role in myelination [90]. High doses of ascorbic acid have inhibitory action on adenylate cyclase activity, the enzyme producing adenyl cyclic adenosine monophosphate (cAMP) [106] and resulted in reduced *PMP22* expression and improved myelination profile, locomotion and survival in C22 mice [69]. However, despite multiple randomized clinical trials [107], ascorbic acid treatment failed to show any significant therapeutic benefit in CMT1A patients [70,71,72,73].

*Histone deacetylase 6* (*HDAC6*) *inhibitor CKD-504* regulates the acetylation of nuclear and cytosolic proteins [108], including heat shock protein 90 (HSP90) and HSP70, which are involved in the folding/refolding of proteins such as PMP22 [80,109]. Testing of CKD-504 enhanced HSP90 acetylation and *HSP70* expression and reduced PMP22 protein levels both in mesenchymal stem cell-derived Schwann cells from CMT1A patients and in the C22 mouse model of CMT1A, in which behavioral, electrophysiological, and histological improvements were demonstrated [80]. Thus, the novel HDAC6 inhibitor, CKD-504 may hold promise for therapeutic efficacy in CMT1A.

*Curcumin* is an anti-oxidant compound that abrogates endoplasmic reticulum (ER) retention and aggregation-induced apoptosis [110]. Oral curcumin was shown to mitigate motor deficits, improve axonal size and decrease neurofilament density in Tr-J mice [81]. In order to overcome the limited pharmacokinetics of curcumin, curcumin-cyclodextin/cellulose nanoparticles were developed (Nano-Cur) [82]. In vitro and in vivo testing of Nano-Cur in the rat model of CMT1A showed reduction of reactive oxygen species (ROS) and improved mitochondrial membrane potential and integrity, leading to improvement of myelination and nerve function [111].

*Melatonin* is another anti-oxidant with anti-inflammatory properties that has been tested as a reducing agent to treat the cellular stress induced by increased production of PMP22 in CMT1A [112]. For this purpose, three children with CMT1A were treated with oral melatonin for 6 months. This treatment reduced oxidation markers and balanced glutathione cycle and pro-inflammatory cytokines levels. Melatonin dietary supplements are FDA-approved and have been extensively used in clinical trials [66,113]. Whether this treatment would benefit nerve pathology in CMT1A remains to be shown.

*Progesterone receptor antagonist.* Progesterone is a neuroactive steroid that indirectly stimulates the promoters of myelin-related genes in SCs [114], including *PMP22*, while its derivatives also trigger myelin gene expression by activating GABA_A_ receptors present in SCs [115]. Subcutaneous injection of onapristone, a progesterone receptor antagonist, in the rat model of CMT1A reduced PMP22 levels, enhanced *Mpz* expression, and improved axonal profiles leading to significant rescue of behavioral abnormalities [83], although myelin thickness was not increased [84]. Moreover, SCs may resist to onapristone treatment by locally synthesizing progesterone [90]. Onapristone clinical trials have never been conducted in CMT1A due to severe side effects observed in cancer patients under this drug [116]. A clinical trial (NCT02600286) was conducted in 2015 with ulipristal acetate (EllaOne^®^), another anti-progesterone drug that is already available in the market, claiming the possibility for EllaOne^®^ to reduce PMP22 levels and oxidative stress in CMT1A patients. However, the trial was terminated due to hepatic side effects, and therefore the potential of anti-progesteron treatment remains uncertain.

*Neuregulin-1 type I* (*NRG1*) is a paracrine growth factor involved in signal exchange cascades between axons and SCs. Under NRG1 dysfunctional conditions neuronal degeneration and abnormal myelin thickness are observed [117]. NRG1 type I (NRG1-I) levels are upregulated in rodent models of CMT1A [59,85], while genetic ablation of SC-derived NRG1-I signaling in C61 mice inhibited hypermyelination and formation of onion bulbs and normalized the misbalanced ErbB2 receptor-MEK/ERK signaling leading to improved phenotype of the model [85].

*Fasting and rapamycin*. Decreased *PMP22* expression, increased production of myelin proteins, as well as improved myelination and locomotor performance was shown in Tr-J mice that were under dietary restriction by intermittent fasting [86]. Since intermittent fasting is not clinically translatable, rapamycin, an FDA-approved calorie restriction mimetic, was introduced into explant cultures of C22 mice and was shown to also improve PMP22 processing and myelin internodal profile, while also enhancing the production of other myelin-related proteins [87]. However, despite these encouraging results in vitro, rapamycin failed to improve neuromuscular performance in vivo in Tr-J model [88].

*Lipid Supplementation.* The rat model of CMT1A exhibits reduced levels of myelin lipids resulting in unbalanced lipids incorporation into myelin. Diet supplementation with phosphatidylcholine and phosphatidylethanolamine improved myelin biosynthesis and nerve function in this model [67]. The lipids supplementation approach is further supported by a recent study showing impairments of sphingolipid and glycerophospholipid metabolism in CMT1A [118]. Although clinical trials showed no side-effects under this treatment [68], it remains unknown whether high doses of dietary phospholipids can benefit CMT1A patients.

*Other molecules.* High throughput in vitro screening of 3000 approved drugs, identified potential compounds to downregulate *PMP22* expression, including bryostatin [119], fenretinide, olvanil and bortezomib [120]. Computational analysis also proposed estradiol as a possible drug to interact with the *PMP22* gene [121]. Increased levels of HSP70 prevented aggregation and enhanced Golgi processing of mutant PMP22 protein in Tr-J mice [122], while inhibition of HSP90 in vitro enhanced chaperone gene expression and improved myelination and processing of PMP22 [123], in line with the effects of HDAC6 inhibitor CKD-504 (above). Both HSP70 enhancer and HSP90 inhibitor molecules are already FDA-approved. However, none of the above mentioned compounds has been further tested in CMT1A clinical trials.

Overall, while the study of CMT1A patients and experimental models in the past three decades has identified several relevant targets for drug therapy based on disease pathophysiology, it remains to be shown whether they could provide translatable and effective treatment approaches for the disease.

### 3.2. CMT1B

CMT1B is the third most common form of demyelinating CMT and is caused by mutations in myelin protein zero (*MPZ*) gene. *MPZ* gene mutations appear to have a variety of cellular effects, resulting in a diversity of phenotypes, ranging from CHN, DSS to CMT1B and CMT2 types. As the main function of MPZ is to act as an adhesion molecule forming tetramers in compact myelin [124], mutations may either impair adhesion properties with dominant effects on the WT protein or may cause misfolding and retention of mutant MPZ in the ER triggering an unfolded protein response (UPR) and apoptosis [125]. CMT1B mouse models used for testing therapies include the S63del (showing the UPR activation) [126], the R98C mutants [127] and the MPZ/P0^+/−^ (P0het) mouse model of heterozygous *MPZ* loss-of-function mutations in humans [128]. Current treatment approaches for CMT1B are mainly focused on drug therapies (Table 2), although more efforts may be also directed in the future towards allele-specific silencing to address the toxic gain of function of *MPZ* mutant alleles.

#### Drug-Based Therapies for CMT1B

*Curcumin.* As in the case of CMT1A, curcumin showed beneficial effects in a CMT1B model, by reducing the accumulation of mutant MPZ in the ER and by reducing apoptosis [110,132]. Daily treatment with curcumin in R98C mice decreased ER stress and UPR, improved motor performance, as well as promoted SC differentiation [130,131]. However, this treatment did not improve nerve conduction velocity or myelination in this model and its potential for clinical efficacy remains unknown.

*Sephin1* is a guanabenz derivative that prolongs the integrated stress response extending the process of restoring proteostasis [129]. IFB-088 is the brand name of Sephin1 and has already been designated by FDA and European Medicines Agency (EMA) as an orphan drug for the treatment of CMT. In 2018 a clinical trial (NCT03610334) tested the safety, tolerability and pharmacokinetic profile of IFB-088 in healthy individuals that were under a daily oral treatment. Following encouraging results of a Phase 1 trial, the results of an ongoing Phase 2 trial of Sephin1 in CMT1B patients is now awaited.

*Neuregulin-1 type III.* Axonal NRG1 type III (NRG1-III) drives myelination by activating transcriptional regulators of myelin genes. Crossing S63del mice with mice overexpressing NRG1-III ameliorated the electrophysiological and morphological deficits as well as increased myelin lipid content [133]. Further preclinical and clinical studies are needed to evaluate the potential of NRG1-III to treat CMT1B pathology.

*Eukaryotic initiation factor 2-phosphorylation and Gadd34.* Eukaryotic initiation factor 2 (eIF2α) is a translation regulator. ER-stress induced by mutant MPZ protein accumulation stimulates the phosphorylation of eIF2α in order to attenuate translation as a rescue mechanism [149,150], while also selectively promoting the translation of anti-oxidant stress response genes [134]. Genetically modified S63del mice with phosphorylation-resistant eIF2α in SCs showed an exacerbated morphological and electrophysiological phenotype, reduction in myelin gene expression, delayed SCs differentiation, but no effect on ER-stress levels. Gadd34 is a phosphatase that de-phosphorylates eIF2α and reactivates protein translation [151]. Genetic or pharmacological limitation of Gadd34 in S63del mice resulted in upregulation of eIF2α phosphorylation and improved Schwann cell differentiation and myelination status, as Gadd34 was found to reactivate translation too aggressively in myelinating Schwann cells [152]. Therefore, either silencing of *Gadd34* or direct enhancement of eIF2α phosphorylation may have beneficial effects in CMT1B.

*Sox2 and Id2* are transcription factors highly expressed by pre-myelinating SCs and then downregulated during myelination [153]. Transcriptomics analysis revealed that nerves of S63del mice express significant levels of pre- and pro-myelinating factors including *Sox2* and *Id2* [152]. Although Sox2 and Id2 are negative regulators of myelination, their sustained expression in CMT1B may have neuroprotective effects [154] and overexpressing their genes may provide a therapeutic benefit for CMT1B.

*Colony stimulating factor 1 receptor (CSF1R) inhibitor (PLX5622)*. CSF1 is macrophage activator expressed by endoneurial fibroblasts mediating macrophage-related neural damage in CMT patients and in mice [155]. Demyelination and disease progression in the P0het model of CMT1B [128] was also mediated by macrophages [156]. Hence, oral administration of the CSF1R inhibitor PLX5622 in P0het mice for 9 months significantly reduced endoneurial macrophages, resulting in amelioration of neuropathological and functional abnormalities in this model [157]. Clinical translation of this approach for CMT1B and other CMT type (discussed below) remains to be optimized.

### 3.3. CMT1X

X-linked type of Charcot-Marie-Tooth (CMT1X) affects 7–15% of all CMT patients and is caused by mutations in the *GJB1* gene [158], encoding the gap junction protein connexin-32 (Cx32) [159,160]. Cx32 is a myelin-related protein expressed specifically in SCs [161] and cell-autonomous loss of its function leads to CMT1X pathology [162,163]. Over 400 different *GJB1* gene mutations have been reported so far (http://hihg.med.miami.edu/code/http/cmt/public_html/index.html#/, last accessed: 27 April 2021) affecting all domains of Cx32, as well as the non-coding gene regions [17,164]. Frameshift, premature stop and non-coding region *GJB1* gene mutations likely cause complete loss of function, whereas missense and in-frame mutations cause intracellular retention of Cx32 [165,166,167,168] in the ER and/or Golgi [167,169,170] with inability to form functional channels. Some mutants may form membrane channels but show abnormal biophysical characteristics [171].

Therapies proposed for CMT1X are mainly focused on gene replacement or delivery of trophic factors to improve peripheral nerve function, or approaches that target the prominent inflammatory component of disease pathology (Table 2). All therapeutic approaches are still in a preclinical phase and employ *Gjb1*-null mice [172] and three different *GJB1* mutant models of CMT1X, the T55I, R75W [163], and N175D [22] transgenic mice.

#### 3.3.1. Gene Therapy for CMT1X

In order to treat CMT1X, the *GJB1* gene needs to be delivered to SCs. This was first attempted using a lentiviral vector carrying the *GJB1* gene driven by the SC-specific *Mpz* promoter [21,162], delivered by direct intraneural injection into the sciatic nerve of 2-motnh old (pre-onset) *Gjb1*-null mice. Intra-sciatic injection restored Cx32 production in paranodal non-compact myelin areas only of the sciatic nerve injected, leading to significant improvement in the ratios of abnormally myelinated fibers/total number of fibers and numbers of foamy macrophages, the main neuropathy features caused by the loss of Cx32 [21].

Although the results of this study were encouraging, intra-sciatic injections cannot be easily translated in patients, and therapeutic benefit can only be expected in the injected nerve. Therefore, lumbar intrathecal injection for PNS vector delivery was developed as a more translatable method [24]. Intrathecal injection of the same lentiviral vector resulted in widespread localization of Cx32 in the paranodal areas of different PNS tissues including lumbar roots, sciatic and femoral nerves. Improvement in behavioral, electrophysiological and morphological analysis was demonstrated in treated 2-month old mice, although phenotype rescue remained partial and did not reach WT levels [24]. Post-onset treatment in 6-month-old *Gjb1*-null mice reproduced the results of early treatment [23] providing further support for this approach in CMT1X patients with mostly advanced neuropathy by early adulthood. Finally, since the majority of CMT1X patients do not lack Cx32 production completely, but rather produce mutant Cx32 that is abnormally retained intracellularly, further studies were undertaken to treat transgenic mice producing representative Cx32 mutant proteins, some of which have shown direct interaction with co-produced WT Cx32 [173,174]. Lentiviral gene therapy showed that virally delivered *GJB1* gene encoding WT Cx32 could overcome the effects of the ER-retained mutant Cx32 protein with T55I substitution, but therapeutic effects were greatly diminished in the presence of Golgi-retained Cx32 mutants with R75W and N175D amino acid residue substitutions [22].

Since treatment with lentiviral vectors could not overcome the effects of some Cx32 mutants and given the limitations of its use due to risk of insertional mutagenesis with systemic in vivo delivery (above), efforts have now focused on developing gene therapy for CMT1X using AAV vectors. Intrathecal injection of AAV9 to deliver the *GJB1* gene driven by the *Mpz* promoter in both pre- and post-onset *Gjb1*-null animals showed expression of *GJB1* in SCs throughout the PNS, at higher levels compared to the lentiviral expression of *GJB1*. Treatment study in the two groups showed both functional and morphological improvements (Figure 1). Two clinically relevant blood biomarkers, elevated neurofilament light (NFL) and the neural cell adhesion molecule-1 (NCAM-1), showed response to treatment as well [34]. It remains to be seen whether the AAV-based approach can be adequately scaled-up in terms of biodistribution to be feasible for clinical application, and whether higher levels of AAV-driven production of WT Cx32 can overcome the interfering effects of certain Golgi-retained Cx32 mutants.

Similarly to CMT1A (above), NT-3 was recently employed as a potential treatment also for CMT1X [33]. NT-3 based gene therapy of CMT1X, resulted in significant improvement of the demyelinating neuropathy in the pre-onset *Gjb1*-null model as indicated by electrophysiological and morphological improvements. Despite this beneficial effect, this approach does not directly solve the cause of the disease and its translation potential remains to be demonstrated.

#### 3.3.2. Targeting Inflammatory Pathways to Treat CMT1X

A potential alternative or complementary therapeutic target for CMT1X is the modification of inflammatory processes that are prominent at least in the mouse model of the disease, in which the presence of foamy macrophages is a characteristic feature of nerve pathology [175]. Several studies have focused on the treatment of CMT1X through amelioration of inflammation. Deletion of the chemokine monocyte chemoattractant protein-1 (MCP-1/CCL2), a mediator of macrophage-related neural damage, reduced the number of macrophages and protected from demyelination in *Gjb1*-null mice [176]. Since MCP-1 is regulated by the ERK pathway, inhibition of this pathway had similar results [176]. Further studies showed that a macrophage-directed CSF-1 mediates the pathology in *Gjb1*-null mice, whereas inhibiting the interaction of CSF-1 with its receptor improved nerve pathology [155,157,177]. Especially secreted proteoglycan CSF-1 (spCSF-1) is related to the macrophage activation and macrophage-related neural damage, while cell surface glycoprotein CSF-1 (csCSF-1) inhibits macrophage activation and attenuates neuropathy [135]. Further studies will be needed to develop clinically useful therapeutics with adequate biodistribution and access to the PNS in order to address this facet of CMT1X pathology.

### 3.4. Other Demyelinating and Recessively Inherited CMT Forms

#### 3.4.1. CMT4B

CMT4B includes three distinct subtypes: CMT4B1 and CMT4B2 are associated with mutations in the myotubularin-related protein (*MTMR*) 2 and 13 genes, respectively, and are characterized by an early onset demyelinating neuropathy with myelin outfoldings [178,179]. CMT4B3 has been associated with mutations in the *MTMR5/SBF1* gene but is characterized by different phenotypes with either a pure demyelinating neuropathy or an axonal polyneuropathy complicated by CNS involvement [180].

Niacin (nicotinic acid) is an FDA-approved drug that decreases cholesterol levels [181] and is known to enhance α-secretase Tace activity, which in turn can downregulate Nrg1 type III signaling that drives myelination. Based on this knowledge, treatment with the niacin extended release formulation Niaspan in *Mtmr2*^−/−^ mice, a model of CMT4B1, resulted in improved myelination and reduction of myelin outfoldings [137].

Myelin growth is also controlled by the Ras-related GTPase Rab35 via complex formation with myotubularins, downregulating lipid-mediated mTORC1 activation. Mice deficient for Rab35 showed hyperactivation of mTORC1 signaling causing myelin overproduction and myelin out-folding similar to *Mtmr2*^−/−^ mice. Since MTMR dysfunction leads to myelin defects by affecting mTORC1, blocking mTORC1 activity could be a therapeutic strategy for CMT4B. Indeed, treatment with the mTORC1 inhibitor rapamycin in Rab35-deficient Rab35^flox/flox^ mice significantly reduced abnormal myelin production [136]. These studies provide proof of principle for translatable therapeutics to treat CMT4B, and future clinical trials are awaited.

#### 3.4.2. CMT4C

CMT4C is an early-onset autosomal recessive form of demyelinating neuropathy and the most frequent among CMT4 types [6], caused by mutations in the *SH3TC2*/*KIAA1985* gene [182]. The localization of SH3TC2 protein in early and late endosomes of the endocytic pathway, in clathrin-coated vesicles in trans-Golgi network, and at the plasma membrane is disrupted when CMT4C causing mutated alleles of *SH3TC2* are expressed in vitro. Moreover, *Sh3tc2*^−/−^ mice develop an early onset progressive hypomyelinating neuropathy with elongated nodes of Ranvier faithfully modeling human pathology [183]. Alterations in the Nrg1/ErbB pathway involved in myelination have been implicated in CMT4C pathogenesis [184], along with impaired neuromuscular junction (NMJ) integrity [185]. CMT4C causing mutations also disrupt the interaction between SH4TC2 and the small GTPase Rab11, a key regulator of recycling endosomes [186,187].

Based on the loss of function mechanism of *SH3TC2* mutations, a gene replacement therapy has been developed for treating the disease and was tested in the *Sh3tc2*^−/−^ model. A lentiviral system was developed to drive the expression of the human *SH3TC2* under the control of the rat *Mpz* promoter. LV-*Mpz.SH3TC2.myc* was delivered into 3-week-old *Sh3tc2*^−/−^ mice by lumbar intrathecal injection and adequate SC-targeted gene expression was achieved. This resulted in behavioral and electrophysiological improvements in treated mice. Moreover, morphological analysis revealed significant reduction in the ratio of the inner to the outer diameter of the myelin sheath (g-ratio) and increase in myelin thickness after treatment along with improved nodal molecular architecture. Finally, elevated blood NFL levels found in the CMT4C mouse model showed amelioration after treatment [25]. A more clinically translatable gene therapy approach for CMT4C remains to be developed in order to facilitate treatment for patients.

#### 3.4.3. CMT4J

CMT4J is caused by recessive, loss-of-function mutations in *FIG4*, encoding the 5-phosphatase of the low abundance signaling phosphoinositide PI(3,5)P_2_ that is localized on the cytoplasmic surface of vesicles of the endosome/lysosome pathway. Nerves expressing the mutated *FIG4* gene are characterized by vacuolization indicative of endosome/lysosome trafficking defects. CMT4J often has a childhood onset and *FIG4* mutations can significantly shorten the lifespan. Gene therapy using AAV9 delivery of a codon-optimized human *FIG4* gene was effective in improving survival rate, motor performance, neurophysiological parameters and morphological abnormalities in the pale tremor mouse (*plt*) model of CMT4J [138]. While intracerebroventricular (ICV) delivery of AAV9-*FIG4* vector at postnatal day (PND) 1 or 4 extended the life span of the *plt* mice from a few weeks to at least a year, delayed treatment at PND 7 or 11 with intrathecal injection resulted in incomplete phenotype rescue, although it still provided benefit. Thus, early AAV9-mediated delivery of *FIG4* is a well-tolerated and efficacious strategy for the disease model, but its translation potential in CMT4J patients remains to be tested.

## 4. Treatment Approaches for Axonal CMT Types

### 4.1. CMT2A

CMT2A is the most prevalent axonal CMT type with a frequency of up to 30% among all CMT2 subtypes and accounts for about 10–15% of all CMT cases [6,188]. CMT2A is caused by mitofusin2 (*MFN2*) gene mutations affecting MFN2, a GTPase protein anchored in the outer mitochondrial membrane, the function of which is mediated by the two transmembrane domains situated close to the C-terminus. Mutations in the *MFN2* gene can be either hereditary (mostly autosomal dominant) or may occur *de novo* [189] and have been shown to result in disruption of normal mitochondrial fusion. This leads to abnormal mitochondrial aggregation and function, along with dysfunctional subcellular mitochondrial trafficking [190,191]. Thus, peripheral nerves consisting of longer axon projections are the most affected in CMT2A patients, possibly due to higher energy demands compared to other cell types.

#### 4.1.1. Targeting the SARM1 Pathway

Axonal degeneration is the hallmark of many neurological disorders, including CMT2A, and is considered to be a genetically encoded program of subcellular self-destruction. The sterile alpha and TIR motif containing 1 (SARM1) protein plays a crucial role in this axonal degeneration program. Intramolecular rearrangement of the molecule leads to dimerization of the TIR motifs and eventually to degeneration of injured axons [192], while SARM1 activation leads to axonal degeneration even in the absence of injury [193]. In vitro studies showed that SARM1^−/−^ primary DRG neurons were resistant to axon degeneration induced by mitochondrial toxicity [194]. Loss of mitochondrial membrane potential leads to reduction of the axon survival factor NMNAT2 which will then activate SARM1 leading to axonal degeneration [195,196]. Thus, in axonal forms of CMT, inhibition of SARM1 may be a promising therapeutic strategy. Since no small molecules are available to inhibit SARM1 action in vivo, a gene therapy approach has been developed using dominant negative SARM1 mutants synergistically packaged into an AAV8 capsid [47]. Intrathecal injection of AAV8-SARM1mutants in WT mice resulted in neuron-specific dominant-negative targeting of SARM1 blocking pathological axon degeneration in vivo. SARM1 inhibition was also demonstrated in vitro by small molecules which inhibit the SARM1 NADase activity, such as isoquinoline [140]. Overall, SARM1 inhibition (along with HDAC6 inhibitors discussed above and below) remains a prime target for treating not only CMT2A but also other axonal and even demyelinating CMT forms with secondary axonal degeneration.

#### 4.1.2. Agonists and Activators of MFN2 Mitochondrial Function

Although there are currently no therapeutics that can directly reverse mitochondrial defects in CMT2A, MFN2 activators have been identified offering a potential therapeutic approach. Novel small-molecule mitofusin agonists were shown to allosterically activate MFN2 reversing the morphological alterations of neuronal mitochondrial defects as well as their impaired mobility caused by two *MFN2* gene mutations in vitro. Moreover, a mitofusin agonist normalized the axonal mitochondrial trafficking in *ex vivo* sciatic nerves of *MFN2* mutant mice [141]. Because these prototype compounds had poor pharmacokinetic properties in vivo, a series of 6-phenylhexanamide derivative mitofusin activators were designed and tested in vivo. This led to the identification of compound 13B as the most promising candidate, as it showed adequate oral bioavailability and increased the number and the motility of mitochondria in the sciatic nerve axons [142].

MFN1 and MFN2 form homo- or hetero- trans-dimers between mitochondria to mediate their fusion. Given the dominant negative effect of mutant MFN2 when the isoform MFN1 is produced at low levels, as it occurs in neurons [139], augmentation of MFN1 or WT MFN2 level may be a useful therapeutic approach to treat CMT2A. Indeed, overexpressing *MFN1* in the nervous system of *MFN2^R94Q^* mutant mice resulted in improved body weight, better behavioral performance and visual acuity, longer survival, as well as reduced mitochondrial aggregation and axon degeneration. Thus, the MFN1/MFN2 ratio appears to be a key factor in neuronal vulnerability to the dominant-negative effects of mutant *MFN2* indicating that increasing the levels of MFN1 or WT MFN2 may be a viable therapeutic strategy to treat CMT2A [139].

### 4.2. Distal Hereditary Motor Neuropathy (dHMN) Associated with SORD Gene Mutations

Distal hereditary motor neuropathy (dHMN) is a recently characterized autosomal recessive disease caused by sorbitol dehydrogenase (*SORD*) gene mutations [144]. *SORD* mutations result in decreased levels and lost function of the SORD enzyme, leading to neuronal sorbitol accumulation, a mechanism previously shown to induce neuropathy in a diabetic mouse model [197]. The polyol pathway in which glucose is converted to sorbitol, has received attention for the treatment of diabetic polyneuropathy and eventually dHMN-SORD [198]. Aldose reductase is a key enzyme in this pathway and its inhibitors have been extensively used in clinical trials. Epalrestat is an aldose reductase inhibitor that was well absorbed into the neural tissue and prevented significantly the decrease of the motor nerve conduction velocity in a diabetic neuropathy model [143]. These beneficial effects in diabetic neuropathy were reproduced with another aldose reductase inhibitor, ranirestat [145]. When epalrestat and ranirestat were used on cultured fibroblasts from patients with *SORD* mutations, elevated sorbitol levels were significantly ameliorated [144], offering a promising approach to treat patients suffering from dHMN-SORD.

### 4.3. CMT2D

CMT2D is caused by dominantly inherited mutations affecting the ubiquitously expressed gene encoding the enzyme glycyl-tRNA synthetase (*GARS*) [199]. Toxic gain-of-function effects of *GARS* mutations has been demonstrated as overexpression of WT *GARS* did not improve the pathological phenotype of two transgenic mouse models of the disease [200]. Thus, suppression of the mutant *GARS* allele should be of therapeutic benefit. Indeed, using allele-specific RNAi delivered by ICV injected AAV9 vector, a dose-dependent therapeutic benefit was achieved in models of CMT2D that persisted for at least a year [31]. However, therapeutic benefit could be achieved only with pre-onset but not with post-onset treatment, and the models used did not express *GARS* mutations causing single amino acid substitutions as found in CMT2D patients, but 5 or 12 base pair changes, making them an easier target for allele-specific RNAi sequences. The feasibility of this strategy or of alternative gene therapy strategies for CMT2D such as allele-nonspecific knockdown combined with gene replacement, allele-specific ASOs, or CRISPR/Cas gene editing remain to be demonstrated at clinically relevant stages of the disease pathology.

### 4.4. CMT2E

CMT2E (allelic to demyelinating CMT1F) is caused by mutations in the *NEFL* gene encoding the NFL subunit [201]. NFL is a major constituent of intermediate filaments and plays a pivotal role in the assembly and maintenance of axonal cytoskeleton. Giant axons are a histological hallmark frequently seen in nerves of patients with CMT2E. Heat shock proteins (HSPs) are involved in the formation of the neurofilament network and in protecting cells from misfolded mutant proteins. Rounding of mitochondria and reduction in axonal diameter occurs before disruption of the neurofilament network, indicating that mitochondrial dysfunction contributes to the pathogenesis of CMT2E. Comparison of neuroprotective effects in a primary motor neuron culture model of CMT2E showed that *HSPA1* and *HSPB1* overexpression prevented neurofilament abnormalities as well as mitochondrial and axonal alterations but their efficacy depended on the specific *NEFL* mutated allele expressed [202]. These findings support the potential use of chaperone-based therapies for the treatment of CMT2E. Induced pluripotent human stem cell (iPSC)-derived motor neurons also provide a useful model to study CMT2E. Using this system, a high-content drug screening platform revealed serine/threonine kinase inhibitors as a potential drug therapy approach for CMT2E [146]. In vivo validation remains to be shown for these therapeutics.

### 4.5. CMT2F

CMT2F is caused by dominantly inherited mutations of the HSP beta-1 (*HSPB1*) gene. HSPB1 is a multifunctional protein with roles in protein aggregation, apoptosis, cytoskeletal maintenance, and gene transcription. Mutations in *HSPB1* result in late-onset dHMN and axonal CMT2F [203]. In vitro cultures of motor neurons carrying the S135F and P182L *HSPB1* gene mutations used as a model of CMT2F, showed a marked decline in the absolute velocity and the percentage of moving mitochondria. These axonal defects were associated with decreased acetylation of α-tubulin. Newly developed HDAC6 inhibitors CHEMICAL X4 and CHEMICAL X9 showed beneficial effects in this model reversing the mutations effects and improving mitochondrial mobility [147]. Furthermore, blocking HDAC6 activity, which is implicated in impaired mitochondrial transport and axon growth inhibition [204], reversed axonal loss and muscle denervation in a mouse model of CMT2F [205]. This approach also benefited CMT2D disease models [206]. Thus, HDAC6 inhibitors may provide a promising therapeutic approach for several forms of CMT2 [204,205,206,207].

### 4.6. CMT2S

Mutations in the immunoglobulin μ-binding protein 2 (*IGHMBP2*) gene cause axonal CMT2S [208] and the more severe, childhood onset motor neuron disease, spinal muscular atrophy with respiratory distress (SMARD1) [209]. The *nmd* mouse model of the disease [210] was employed to test the efficacy of AAV9-*CBA*-*IGHMBP2* intravenous injection at postnatal day 1 (P1) as a therapeutic approach to treat CMT2S and SMARD1 [45]. This treatment showed high efficacy with amelioration of motor dysfunction and muscle degeneration. Moreover, in another treatment trial using *nmd* mice, ICV injection resulted in significantly improved muscle pathology and prevented motor neuron loss. However, optimal dosing remains a crucial parameter to be addressed as overexpression of IGHMBP2 may also have a detrimental effect [148].

Overall, several promising therapeutics are on the horizon for axonal CMT types, including some that target specifically the disease pathogenesis, and others that target pathways of mitochondrial dysfunction and axonal degeneration, that may be applicable to most if not all CMT types.

## 5. Clinical Trial Readiness for CMT Neuropathies

While there is a multitude of emerging treatments currently studied for CMT neuropathies, they remain mostly at the pre-clinical level and their clinical translation creates major challenges ahead. As outlined above, clinical trials have mostly been conducted for the treatment of the most common type, CMT1A, often with discouraging results. One of the major hurdles was the unavailability of sensitive outcome measures that would allow the detection of treatment response in a slowly progressive chronic disease such as CMT1A, which also shows marked phenotypic variability. These issues are relevant to other CMT neuropathies as well, but in addition, due to the fact that they are much more rare compared to CMT1A, or even ultra-rare, additional challenges of special clinical trial design will have to be overcome in order to achieve successful clinical application of promising experimental therapeutics. Below we discuss the need for more accurate outcome measures, novel biomarkers, and for optimal clinical trial design, as they apply to most CMT neuropathies.

### 5.1. Clinical Evaluation Tools

Efforts over the last 10 years have focused on establishing clinical trial readiness for CMT neuropathies. These studies were also motivated by the failures of initial trials in CMT1A. Two composite scoring systems have been developed incorporating symptoms and neurological exam, the Rasch modified CMT Neuropathy Score (CMTNS-R) and the Rasch modified CMT Examination Score (CMTES-R) [211]. The CMTNS is the only validated tool for measuring progression in CMT [212]. However, CMTNS proved not to be sensitive enough for detecting meaningful changes within the study period as it only showed small annual progression in clinical trials [72,73]. Thus, at least several hundred patients would be needed in a double blinded trial to detect significant slowing of disease progression [213]. Moreover, improvements in CMTNS score progression were even observed in the placebo groups [72,214] unlike the previously available natural history data [215]. Thus, careful choice of interpretable outcome measures and large enough participant numbers to ensure statistical power will be needed, as demonstrated by the ascorbic acid and PXT3003 trials [105]. Of importance is also to use functional outcome measures in addition to specific readouts that are inherent to the specific treatment approach, such as the increase in muscle bulk and assessment of foot dorsiflexion in the ACE-083 trial (NCT03124459).

Functional and patient-reported outcome (PRO) measures are additional tools currently being introduced to improve clinical trial design in CMT neuropathies. The CMT Functional Outcome Measure (CMT-FOM) is a modification of CMT Paediatric Scale (CMTPedS) [216] aiming at providing a valid, responsive and meaningful instrument of function for adults (>18 years old). It is a 13-item assessment that measures physical ability in adults with CMT and has already been validated for CMT1A [217]. The CMT-FOM discriminated between participants with clinically mild as opposed to moderate-severe CMT1A and was sensitive enough to show functional limitations in patients with the mildest CMT1A who demonstrated a floor effect on the CMTES. CMT-FOM has shown excellent inter-rater reliability [218], however longitudinal studies will be necessary to determine its responsiveness and utility for clinical trials.

The CMT Health Index (CMT-HI) is a PRO to measure the burden of CMT [219] and will also be required in future trials to demonstrate responsiveness to disease progression. CMT-HI contains 18 themes to capture disease burden with a high internal consistency and test-retest reliability, that was able to discriminate between patient groups with different disease burden as well as between levels of disability as measured by the CMTES and the mobility-Disability Severity Index [220].

Finally, wearable sensors are being tested as additional tools to measure daily activity outside the clinic as well as to evaluate gait and balance during the 6-min walk test (6MWT), timed up and go (TUG), and the 10-m walk of the CMT-FOM [221,222]. The 6MWT highly correlated with all previously used outcome measures. The StepWatch(™) Activity Monitoring correlated only partially with commonly used outcome measures, but nevertheless provided data that correlate better with quality of life measures [222].

### 5.2. MRI and Other Biomarkers

Progression of CMT neuropathies results in characteristic muscle atrophy and fat replacement of muscle tissue, providing a relevant outcome measure that can be directly assessed by neuroimaging. MRI imaging of lower limbs has been mainly studied in CMT1A patients, demonstrating that the free intramuscular fat accumulation (IMFA) increases by 1–2% per year within calf muscles independently of a subject’s overall level of activity or fitness, making it the most sensitive measure of progression that has been identified to date [223,224,225]. Evaluation of both thigh and calf muscles will enhance the MRI sensitivity across different stages of the disease, as proximal thigh muscle involvement may continue to progress in a linear mode even after the distal calf muscle have reached an end-stage condition reducing their IMFA sensitivity to detect further changes. Thus, adding the MRI for IMFA assessment of both proximal and distal limb muscles as a primary outcome measure in clinical trials will significantly improve the chance of detecting treatment response even with lower participant numbers in CMT1A and other more rare CMT forms.

Blood biomarkers provide another emerging tool that may complement clinical and MRI-based outcome measures. Molecular markers reflecting abnormalities in SCs, degenerating axons, or denervated muscle could serve as biomarkers for severity and progression in CMT neuropathies. Recent studies have shown increased plasma NFL levels in patients with different types of CMT [226]. Furthermore, plasma NFL elevation was shown to correlate with disease severity measured by the CMTES or by the CMTNS, as well as with age [226]. NFL levels were also found to be elevated in mouse models of CMT1X [23,34] and CMT4C [25]. Importantly, amelioration was demonstrated in animals treated with gene therapy, indicating that this may be a treatment-responsive biomarker as well.

The transmembrane protease serine 5 (TMPRSS5) protein is highly produced in SCs [227] and was found to be significantly elevated in the plasma of CMT1A patients by 2.07-fold in two independent cohorts of CMT1A patient samples. However, TMPRSS5 level was not correlated with disease score (CMTES-R, CMTNS-R), nerve conduction velocities (Ulnar CMAP, Ulnar MNCV), or with age, and thus may be less useful as a clinical outcome measure.

Further blood biomarkers are currently investigated using proteomics approaches. Significant serum elevation of neural cell adhesion molecule-1 (NCAM1) was shown in CMT1A patients as well as in patients with various forms of inflammatory neuropathies [228]. This biomarker was also elevated in a mouse model of CMT1X and showed responsiveness to *GJB1* gene replacement therapy [34]. NCAM-1 was shown to regulate synaptic reorganization after peripheral nerve injury suggesting an important role during regeneration [229]. Thus, it may provide, along with NFL, a valuable marker of CMT pathology and should be considered for inclusion in future trial design. However, the natural history of progression of all these candidate blood biomarkers and how they correlate with other CMT outcome measures remains to be demonstrated in further longitudinal studies.

### 5.3. Optimal Trial Design for CMT Neuropathies

Horizontal aspects relevant to all CMT types are the establishment of longitudinal natural history data, which currently exist only for CMT1A, and the optimal clinical trial design tailored to the rarity of the specific CMT type [230]. Innovative clinical trial design will have to incorporate FDA requirements, while availability of longitudinal natural history data will allow the design of clinical trials suitable for rare and ultra-rare CMT types, such as the seamless phase 1/2/3 trial encompassing dose escalation and safety, and the group sequential design [231]. Multi-center trials will be needed and open label design may be necessary for very rare types. The initial clinical trials are likely to include only adults, while the inclusion of children may be considered from regulatory point of view only in CMT types with severe and early phenotypes. Finally, we need to choose therapies with strong chances for success based on compelling biological evidence, as the number of trials that can be conducted is limited and failures can slow down an otherwise dynamic and promising path towards CMT treatments.

## 6. Summary and Future Perspectives

The field of inherited neuropathies has progressed a long way from the clinicopathological era of the nineteenth century and the initial description of the disease by Charcot, Marie, and Tooth, with only supportive treatments, to the molecular genetic era in the last 30 years, and very recently entering an exciting new era of emerging treatments. In this review we highlight how the diverse mechanisms of CMT neuropathies require a personalized medicine approach with disease-specific therapies, while final common pathways of axonal degeneration are also a major therapeutic target. This is a very active field with novel treatment approaches for several CMT forms currently in preclinical and in clinical trial stage. Recent successful clinical translation of novel therapeutics in other neuromuscular disorders, including gene therapy, has provided further impetus to develop effective therapies for CMT patients as well. Lessons learned from the initial clinical trials in CMT1A have stimulated the development of more comprehensive clinical evaluation tools, the discovery of sensitive disease biomarkers that could be treatment-responsive, and the optimization of clinical trial design. Thus, while many challenges remain in the path towards cures for CMT neuropathies, we have reasons to be hopeful for successful treatments in several CMT types in the near future.

## Figures and Tables

**Figure 1 ijms-22-06048-f001:**
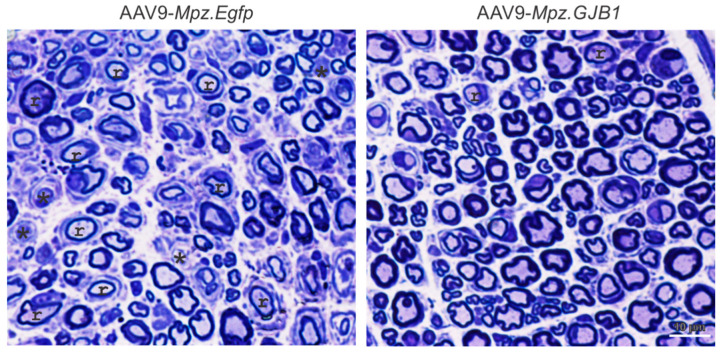
Effects of AAV9-mediated *GJB1* gene replacement therapy in femoral motor nerves of the *Gjb1*-null mouse model of CMT1X. Toluidine blue stained femoral motor nerve semithin sections from 10-month old *Gjb1*-null mice treated post-onset at 6 months of age with intrathecal delivered mock vector (AAV9-*Mpz.Egfp*, left) or with AAV9-*Mpz.GJB1* vector (right) in Schwann cells, shows improvement of myelination defects including the remyelinated (r) and demyelinated (asterisk) fibers. Modified from [34].

**Table 1 ijms-22-06048-t001:** Clinical and pre-clinical stage emerging treatments for CMT1A.

Compound	Rationale	Evaluation Stage	References
CLINICAL TRIALS			
Gene Therapy			
AAV1 delivered *NT-3*	Neurotrophic activity	Phase I/II Clinical trial	[29,43] (NCT03520751)
*Drug Therapy*			
PXT3003: cocktail of baclofen, naltrexone and sorbitol	Downregulation of *PMP22* overexpression	Phase III Clinical trial	[63,64,65] (NCT04762758)
Melatonin	Anti-oxidant activity	Pilot clinical trial	[66]
Dietary lipid supplementation	Stimulates myelin biosynthesis	Pilot clinical trial	[67,68]
Ascorbic acid	Inhibition of cAMP pathway downregulates *PMP22* overexpression	Phase II Clinical trial/Unsuccessful	[69,70,71,72,73] (NCT00484510)
Progesterone receptor antagonist (EllaOne^®^)	Anti-progesterone activity, inhibition of myelin-related genes expression in SCs	Phase II Clinical trial/Unsuccessful	(NCT02600286)
PRE-CLINICAL STUDIES			
Gene Therapy			
VM202: novel genomic HGF cDNA hybrid	Stimulation of SC repair and regeneration		[74,75]
siRNA	Allele specific downregulation of *PMP22* overexpression		[76]
siRNA	Downregulation of *P2RX7* overexpression to reduce abnormal Ca^2+^ influx into SC		[77]
siRNA conjugated to squalenoyl nanoparticles	Downregulation of *PMP22* overexpression		[44]
shRNA	Downregulation of *PMP22* overexpression		[42]
Lentiviral delivered miR-318	Overexpression of miR-318 downregulates overexpressed *PMP22*		[26]
AAV2 delivered miR-29a	Overexpression of miR-29a downregulates overexpressed *PMP22*		[30]
ASOs	Downregulation of *PMP22* overexpression		[78]
Antiparallel triplex-forming oligonucleotides	Bind on *PMP22* promoters to downregulate overexpressed *PMP22*		[79]
CRISPR/Cas9	Deletion of TATA-box of *PMP22* gene promoter in order to downregulate *PMP22* overexpression		[41]
Drug Therapy			
P2X7 Inhibitor	Downregulation of *P2RX7* overexpression in order to reduce abnormal Ca^2+^ influx into SC		[77]
CKD-504	HDAC6 inhibitor, downregulation of *PMP22* overexpression		[80]
Nano-Cur	Modified curcumin, anti-oxidant activity		[81,82]
Progesterone receptor antagonist (Onapristone)	Inhibition of SCs myelin-related genes expression		[83,84]
NRG1	Paracrine growth factor, genetic ablation		[85]
Fasting and rapamycin	Improve ER processing of overproduced PMP22		[86,87,88]
Upregulation of c-Jun transcription factor	Stimulates myelin gene expression		[89,90]

**Table 2 ijms-22-06048-t002:** Summary of emerging treatment approaches for other CMT neuropathies.

Compound	Rationale	References
Demyelinating CMT neuropathies		
CMT1B		
CLINICAL TRIALS		
*Gene Therapy*		
IFB-088: *Sephin1*	Proteostasis restoring, clinical testing for safety	[129], (NCT03610334)
*PRE-CLINICAL STUDIES*		
*Drug Therapy*		
Curcumin	Anti-oxidant activity	[110,130,131,132]
NRG1-III	Activates transcriptional regulators of myelin genes	[133]
Genetic inactivation of *Gadd34*	Upregulation of eIF2α phosphorylation controlling translation	[134]
CMT1X		
Gene Therapy		
Lentiviral delivered *GJB1* gene	Schwann cell specific Cx32 production	[21,24]
AAV9 delivered *GJB1* gene	Schwann cell specific Cx32 production	[34]
AAV1 delivered *NT3*	Neurotrophic factor expression	[33]
*Drug Therapy*		
CSF1 receptor antagonists	Blocking CSF-1 effects reduces inflammation in CMT1X model	[135]
CMT4B		
Drug Therapy		
Rapamycin	mTORC1 inhibitor restores Rab35 regulatory role on myelin production	[136]
Niaspan	Enhancing Tace activity to downregulate Nrg1 type III signaling	[137]
CMT4C		
Gene Therapy		
Lentiviral vector delivered *SH3TC2*	Schwann cell specific *SH3TC2* gene expression	[25]
CMT4J		
Gene Therapy		
AAV9 delivered *FIG4* gene	Restoration of *FIG4* expression	[138]
Axonal CMT neuropathies		
CMT2A		
Gene Therapy		
AAV8 delivered *SARM1* mutants	Dominant negative mutants block the wild type *SARM1* function	[47]
*MFN1* genetic addition	Compensates mutated *MFN2* dysfunction	[139]
Drug Therapies		
Isoquinoline	Inhibition the SARM1 NADase activity	[140]
MFN2 agonists	Improvement of mitochondrial trafficking	[141]
6-phenylhexanamide derivative mitofusin activators	Improvement of mitochondrial motility	[142]
dHMN-SORD		
Drug Therapies		
Epalrestat	Aldose reductase inhibitor that normalizes abnormal sorbitol levels	[143,144]
Ranirestat	Aldose reductase inhibitor that normalizes abnormal sorbitol levels	[144,145]
CMT2D		
Gene Therapy		
AAV9-delivered artificialmiRNA	Allele-specific knockdown of dominant *GARS* mutants	[31]
CMT2E		
Drug Therapies		
Serine/threonine kinase inhibitors	Partially reverse neurofilament deposits phenotype in motor neuron axons	[146]
CMT2F		
Drug Therapies		
HDAC6 inhibitors	Restored acetylated α-tubulin levels improving mitochondrial mobility	[147]
CMT2S		
Gene Therapy		
AAV9-delivered *IGHMBP2* gene	Restoration of *IGHMBP2* gene function	[45,148]

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
