# Peer review of "Emerging Therapies for Charcot-Marie-Tooth Inherited Neuropathies"

_ijms, 2021, doi:10.3390/ijms22116048_

Round 1

Reviewer 1 Report

The authors in their review “Emerging Therapies for Charcot-Marie-Tooth Inherited Neuropathies” give an excellent account of the research of new therapies for Charcot-Marie-Tooth neuropathies. The review introduces the different types of CMTs including their clinical features as well as their classification and related genetics. It also details various approaches that have been used so far to treat these neuropathies discussing their limitations and relative advantages.

The review reads generally well, is of potential interest to a broad audience of MDs and scientists. It is organised in a logical way and is referenced correctly.

I have very few comments that I hope could help in improving the outlay of the manuscript:

  • There are barely no typos, the only ones found:

 Line 124 would be: …lentiviral incorporation into the hosts genome..

Lines 577 and 578: in vivo would be written in italics “in vivo” like the rest of the manuscript

  • Think that the tables outlining tools would benefit from a bit of colour maybe grouping different headings into different colours
  • I guess authors have chosen Figure 1 to refer to the latest publication of Kagiava et al., might need to refer to in the figure legend.

Author Response

We would like to thank the reviewers for very helpful and insightful comments. We made every effort to address them and highlight the respective changes in the manuscript.

REVIEWER 1:

Comments and Suggestions for Authors

The authors in their review “Emerging Therapies for Charcot-Marie-Tooth Inherited Neuropathies” give an excellent account of the research of new therapies for Charcot-Marie-Tooth neuropathies. The review introduces the different types of CMTs including their clinical features as well as their classification and related genetics. It also details various approaches that have been used so far to treat these neuropathies discussing their limitations and relative advantages.

The review reads generally well, is of potential interest to a broad audience of MDs and scientists. It is organised in a logical way and is referenced correctly.

 I have very few comments that I hope could help in improving the outlay of the manuscript:

There are barely no typos, the only ones found:

  • Line 124 would be: …lentiviral incorporation into the hosts genome.

Incorporated

  • Lines 577 and 578: in vivo would be written in italics “in vivo” like the rest of the manuscript

Incorporated

  • Think that the tables outlining tools would benefit from a bit of colour maybe grouping different headings into different colours

Incorporated

  • I guess authors have chosen Figure 1 to refer to the latest publication of Kagiava et al., might need to refer to in the figure legend.

Incorporated

Reviewer 2 Report

Please find attached document with comments

Author Response

REVIEWER 2:

The opinion about the manuscript by Marina Stavrou , Irene Sargiannidou , Elena Georgiou , Alexia

Kagiava , Kleopas A Kleopa entitled “Emerging Therapies for Charcot-Marie-Tooth Inherited Neuropathies”  

First we would like to extend special thanks to this reviewer who has so kindly and generously spent considerable amount of time to go through the manuscript in detail and provide extensive suggestions for improvement- we really appreciate it and tried as best we could to follow most of the suggestions.

General comments

In this review all achievements in the experimental therapy in the area of CMT diseases have been addressed with great precision. The corresponding author of this paper has 20 years long experience in CMT research and world-recognized achievements in the gene therapy research. This paper seems to cover almost all aspects of therapy. However, in my opinion, it does not sufficiently address the area of possible anti-inflammatory treatment in CMT. Especially the studies concerning MPZ, GJB1 and PMP22 knock-out mice should be presented.

The authors do not indicate the leading direction in this area of research.

RESPONSE: The developments around anti-inflammatory therapies in CMT1X (GJB1) related models had already been described in detail in section 3.3.2 of our initially submitted manuscript. We have now added additional references about anti-inflammatory studies in the CMT1B model. We had not included PMP22 knock-outs or HNPP in our review and we could not find evidence of effective anti-inflammatory treatment in the CMT1A model.

The authors should even, in more general terms, ask the question concerning the lack of CMT- drugs of the market. For comparison in a short time in SMA the third drug is considered to be introduced to therapy. What are the main factors hampering research in CMT-therapy?

RESPONSE: This issue has been addressed in Section 5, including the problems with slow disease progression, inadequate design of previous clinical trials, and the need to develop more sensitive outcome measures.  We added points on this in the introductory section 1.

The authors characterize the main CMT-scores used in the clinic but they do not show their limits in the certain group of patients (children, patients with atypical symptoms).

RESPONSE: Going into further detail about clinical outcome measures and in particular the main CMT scores we believe is beyond the scope of the current review. This review is focused on emerging therapies, and not on clinical trial design or clinical CMT scores. Other excellent reviews on this topic and original publications have been cited.

To conclude this paper will be more interesting when rewritten from the aggregation of data concerning various approaches in CMT therapy to a more personal view of the authors.

RESPONSE: We have provided our personal views in several of the treatments discussed (Highlighted in the text), however we believe that this review should mostly provide a comprehensive and objective overview of the field based on the available literature, and should not reflect a personal view of the authors for or against certain treatment approaches.

Detailed comments

In several places the colloquialisms are used which made the text unprecise. There is written several times in a text “ameliorating of phenotype”, “improved the phenotype” (Lines (L.) 133, 197, 262, 276, 303, 316, 317, 327-328, 335, 390) or “phenotype improvement”, “phenotype rescue” (L. 434, 585), “rescuing the phenotype” (L. 529)– what do these statements really mean? 

Moreover, the statements like – improved pathology, benefit nerve pathology etc. are used in terms of reduction of pathological symptoms, but logically, the statements actually mean, for me, deterioration of the condition, not improvement, as the authors wanted to say. L. 224 (ameliorating nerve pathology), 378 (ameliorated […] deficits), 457 (improvement of the demyelinating neuropathy), 466 (amelioration of inflammation), 473 and 661 (improved muscle pathology), 521 (amelioration after treatment). Please correct to have a proper meaning.

We do not follow the reviewer’s point here. Amelioration is a widely used term in scientific literature and not a “colloquialism”. It means to make (something bad or unsatisfactory) better. For example, in the new reference added in CMT1B section based on the reviewer’s suggestion above (Nr. 146, Klein et al., Brain 2015), in figure 4 the title states: “Figure 4 Long-term CSF1R inhibition ameliorates neuropathic features in P0het but not in PMP22tg mutants.” So we did not invent the term amelioration. And how could “improved pathology, benefit nerve pathology etc.” mean “deterioration of the condition, not improvement” is not obvious to us. In any case, we tried to be as specific as possible and we adjusted where necessary the text accordingly

Italic of genes names is missing in L. 209, 211, 213, 214, 218 (x2), 219, 221, 257, 275, 283, 300, 317, 340, 342, 447 (MPZ promotor), 515, 585 (x2), 632, 641 and in tables

Adjusted accordingly

FDA abbreviation is not introduced – should be when for the first time it appears in line 249; Abbreviation EMA is used once in line 371 without explanation.

Adjusted accordingly

The genes are expressed, deleted and mutated, the proteins are absent, produced but in a text these concepts are mixed. The proposed corrections are listed below with other required ones.

The model could not be of one of the types from classification. It could be only a model of disease classified as CMTxx type  (L. 184, 261, 315, 325, 414, 479, 586, 611, 629, 635, 648, 749).

Similarly, there are no CMTxx mutants, there are, for example, GARS, GJB1, PMP22, MPZ etc. mutants.

Also, the classification type CMTxx cannot be caused by a mutation. CMTxx type disease is caused by mutations in the xxx gene. Please correct.

Thank you for the suggestions, we clarified the above terms where necessary throughout the text.

  1. 8 Inherited – not in bold

Adjusted accordingly

  1. 81 and 93 remove recent – not all publications are recent – Is publication from 1993 recent?

Removed

  1. 87 colloquialism: based on the causative genes – based on the mutations in specific genes

Adjusted accordingly

  1. 88-90 colloquialism: mutations in Schwann cell(SC)-associated genes typically cause demyelinating neuropathy, whereas mutations affecting neuro-axonal genes – please re-write

We do not see a “colloquialism” in this sentence

  1. 114 Should be absence of protein, and the wild type version of the protein could not be added. Only the wt gene could be introduced what results in production of wt protein. Please re-write.

Adjusted accordingly

  1. 115 Should be: In types caused by increased gene dosage effect

Adjusted accordingly

  1. 117 Lentiviral vectors [...] efficient delivery (of what?) and their therapeutic…– please re-write this sentence, because at present form the sentence can be understand that vectors per se are therapeutic agents. Adjusted accordingly
  2. 122 Similarly to L.117 expression level of what?

Adjusted accordingly

  1. 126 What is episomal integration of AAV?

We corrected to episomal persistence

  1. 141-144 The clustered regularly interspaced short palindromic repeats (CRISPR) working in cocooperation with Cas genes in order to disrupt, delete, add or replace a sequence of the targeted DNA or RNA regions are not a method the method is their usage. Please correct

Adjusted accordingly

  1. 149 Local administration of what? Explained accordingly
  2. 154 intravenous delivery of what? Explained accordingly
  3. 158 applicable delivery method – delivery of what? Viruses per se are not therapeutic

Corrected accordingly

  1. 209 Should be: and reduced PMP22 protein level Adjusted accordingly
  2. 220-222 I don’t understand the sentence “CRISP/Cas9….” – please make it clear Adjusted accordingly
  3. 229 overproduced PMP22, causing Ca2+ influx (from where?) Explained accordingly
  4. 230 I don’t understand co-culture of what types of cells. Explained accordingly
  5. 274 ascorbic acid could not have an inhibitory action on cAMP but it has on the adenylate cyclase activity, the enzyme producing cAMP – re-write Adjusted accordingly
  6. 290 improved mitochondrial membrane – what does it mean? Explained accordingly
  7. 293 the patients’ status cannot be hyperoxidative - the oxidative stress in patients’ cells?

Adjusted accordingly

  1. 306 as severe side effects were observed to develop in cancer patients under this drug

Adjusted accordingly

  1. 322 production of other Adjusted accordingly
  2. 327 model animals Adjusted accordingly
  3. 333 physical structure of what? Adjusted accordingly
  4. 337 produced in SCs Adjusted accordingly
  5. 342 (HSC70) encoding gene Adjusted accordingly
  6. 344 chaperones genes expression Adjusted accordingly
  7. 352 MPZ gene mutations Adjusted accordingly
  8. 356 WT protein or Adjusted accordingly
  9. 382 mutant MPZ protein accumulation Adjusted accordingly
  10. 404 GJB1 gene mutations Adjusted accordingly
  11. 407 What are non-coding mutations? Explained accordingly
  12. 409 Missing the advance introduction of the Cx32 proteins that form the channels

We do not understand the meaning of this change, and therefore cannot change the respective sentence, which is clear as it is in our view.

  1. 423 restored Cx32 production (or GJB1 expression) Adjusted accordingly
  2. 437 do not lack Cx32 production (or GJB1 expression) Adjusted accordingly
  3. 438 rather produce mutant Cx32 Adjusted accordingly
  4. 439 mutations are not expressed the expressed is mutated GJB1 gene Corrected accordingly
  5. 440 co-produced We clarified gene expression
  6. 441 Cx32 is not delivered, delivered is DNA fragment with GJB1 gene encoding Cx32 protein – please re-write Adjusted accordingly
  7. 442 The mutant – organism/protein not the type of disease – retained mutant Cx32

Adjusted accordingly

  1. 443 retained Cx32 mutants with N175D and R75W amino acid residue substitutions; the order should be: first R75W and next N175D as it is order of these residues in a protein Adjusted accordingly
  2. 446 therapy efforts are not concentrated on AAV vectors but on developing gene therapy using AAV viruses Adjusted accordingly
  3. 448 expression of GJB1 in SC Adjusted accordingly
  4. 449 level of Cx32 […] the effect of lentiviral delivery (of what?) of the same gene. We clarified
  5. 454 levels of expression of virally delivered GJB1 gene Adjusted accordingly
  6. 455 there are no CMTX1 mutants, there may be GJB1 mutations or mutant Cx32 protein Adjusted accordingly
  7. 487 delete word protein The word “protein” belongs to the name of this gene and will not be deleted (Mtmr2 = myotubularin-related protein 2)
  8. 473-474 The terms spCSF-1 and csCSF-1 are not explained, what are sp and cs stand for?

Explained accordingly

  1. 492 Myotubularins related proteins being the phosphatidylinositol-3-phosphate phosphatases

Adjusted accordingly

  1. 499 In both cells – what cells, what does it refer to? deleted
  2. 504 Localization of SH3TC2 protein in early and late endosomes of the endocytic pathway

Adjusted accordingly

  1. 506 when mutated alleles of SH3TC2 gene, causing CMT4C, are Adjusted accordingly
  2. 510 CMT4C causing mutations Adjusted accordingly
  3. 512 endosomes Adjusted accordingly
  4. 515 under the control of MPZ promotor this is rat Mpz (not human promoter (and not promotor)
  5. 519 explain what is g-ratio Explained accordingly
  6. 521 hallmark is of disease not of model deleted
  7. 525 Not all nerves are not vacuolized only with mutated FIG4 gene Adjusted accordingly
  8. 544 shown to result in disruption of normal… Adjusted accordingly
  9. 574 in vitro Corrected accordingly
  10. 582 to mediate fusion of what? Explained accordingly
  11. 584 MFN2 level may Adjusted accordingly
  12. 586 I don’t understand tissue vulnerability to what? Explained accordingly
  13. 587 increased level of MFN1 or wt MFN2 protein Adjusted accordingly
  14. 629-633 The sentence is too long and the main message gets lost. Does it really matter here that the HSPs mentioned are regulated by Hsf1? Adjusted accordingly
  15. 642-643 There is no cultures of mutation. Motor neurons carrying mutations in the HSPB1 gene, resulting in substitutions S135F and P182L, from in vitro cultures, used as a model of CMT2F disease, showed a … Corrected accordingly
  16. 645 There is a logical gap between two sentences. Corrected accordingly
  17. 649 this model – which one?, two are mentioned above Corrected accordingly
  18. 664, 679 Is it readiness the proper word here? measurable outcome, indicator? Yes, we find this a proper word because it encompasses all of the aspects that are important for being ready to conduct successful clinical trials in CMT neuropathies.
  19. 678 tools Corrected accordingly
  20. 688 The improvement in scale progression is not of interest to anyone. We disagree. The observed improvement in the placebo group has a lot of relevance for how to plan future clinical trials in a very slowly progressive disease and also has contributed to failure to show efficacy.
  21. 690 What does “powering of trials” mean? Clarified
  22. 692 What does “using functional outcome measures” mean? Already explained in paragraph below
  23. 696 The one clinical trial? Not sure what is asked here- we talk about clinical trial design in general
  24. 740 highly produced Adjusted accordingly
  25. 742 TMPRSS5 level was Corrected accordingly
  26. 757 What does “horizontal aspects” mean? Explained in the same sentence: relevant to all CMT types

Chapters 3.3 and 3.3.1 Information about the Cx32 protein is separated in these parts of the text, reorganize the text to clearly present the information without redundancy, repetition.  Adjusted accordingly

TABLES

Table 1

  1. Names of genes in italic Adjusted accordingly
  2. In column Rationale please uniform type of information and form of sentence

Adjusted where necessary

  1. in line fasting and rapamycin - Improved ER processing of overproduced PMP22

Corrected accordingly

Table 2

  1. Names of genes in italic Corrected accordingly
  2. In column Rationale please uniform type of information and form of sentence Corrected accordingly
  3. CMT4B move right Corrected accordingly
  4. It can’t be “restores mitochondrial dysfunction” – illogical Corrected accordingly
  5. dHMN-SORD The improvement of the model? Is it really about creating a better model? Corrected accordingly
  6. CMT2D mutation can be dominant not a mutant – dominant mutation in GARS gene

Corrected accordingly

  1. CMT2S Improvement in pathology? Corrected accordingly

Reviewer 3 Report

Authors reviewed in detail clinical and pre-clinical studies based on therapeutic strategies in Charcot-Marie-Tooth disease. Moreover, authors examined the major problem of outcome measures lacking, that still represents the most important obstacle in therapeutic trials. The review is extensive and well written, even though the structure of each paragraph is quite monotonous. I have some few comments:

  • Authors should include a brief methodological section in which clarify how they found papers and which is the years they included in the research.
  • Tables should be more informative to help reading of the main text. For example, over the action mechanisms, in the table should be included model type (e.g., rat, mouse, Drosophila) and major outcome (e.g., improving of electrophysiology, pathology or motor function, normalizing protein expression).
  • Authors should cite the stepwatch activity monitor (doi: 10.1111/ene.13033) as wearable sensors.

Author Response

REVIEWER 3:

Authors reviewed in detail clinical and pre-clinical studies based on therapeutic strategies in Charcot-Marie-Tooth disease. Moreover, authors examined the major problem of outcome measures lacking, that still represents the most important obstacle in therapeutic trials. The review is extensive and well written, even though the structure of each paragraph is quite monotonous. I have some few comments:

  • Authors should include a brief methodological section in which clarify how they found papers and which is the years they included in the research.

We have not construct this review as a systemic review. We followed the journal’s guideline that references should be up to-date, i.e., with 50% or more of cited papers published within recent 5 years

  • Tables should be more informative to help reading of the main text. For example, over the action mechanisms, in the table should be included model type (e.g., rat, mouse, Drosophila) and major outcome (e.g., improving of electrophysiology, pathology or motor function, normalizing protein expression).

We believe that adding all this information in the tables will make them too busy and will make information in the text redundant.

  • Authors should cite the stepwatch activity monitor (doi: 10.1111/ene.13033) as wearable sensors.
  • We already cited this reference and have expanded the description.

Round 2

Reviewer 2 Report

The comments are in an attached file.

Author Response

RESPONSE TO REVIEWER’S COMMENTS

We would like to thank reviewer 2 for additional helpful comments. We made every effort to address them and track the respective changes in the manuscript.

The manuscript by Marina Stavrou , Irene Sargiannidou , Elena Georgiou , Alexia Kagiava , Kleopas A Kleopa entitled “Emerging Therapies for Charcot-Marie-Tooth Inherited Neuropathies” is improved and deserves publication in IJMS, but still minor changes are required. I haven’t noticed them during first round of reviewing.

Detailed comments

  1. 151, 155, 158, 164, 521, 546 bio-distribution or biodistribution – the form must be unified in a manuscript

Unified so that biodistribution is used throughout the manuscript

There is still no distinction between genes and proteins in some places and some genes names are not in italic.

  1. 209 protein level Adjusted accordingly
  2. 240 overproduced PMP22 - P2X7 could only interact with PMP22 protein Adjusted accordingly
  3. 254 NTF3 cDNA encoding NT-3 Adjusted accordingly
  4. 263 contacted ? maybe conducted Adjusted accordingly
  5. 270 – 273 I don’t understand the sentence after changes. I think it should be: As in gene therapy approaches, drug therapies for CMT1A are intended to reduce the toxic effects of overexpressed PMP22. Adjusted accordingly
  6. 278 pathways Adjusted accordingly
  7. 354 increased production of Adjusted accordingly
  8. 368 glycerophospholipid synthesis/degradation (?) pathways Metabolism is the word used in the original paper, so we adjusted accordingly
  9. 383 interact with the mutant PMP22 Adjusted accordingly
  10. 404 (showing the UPR activation) Adjusted accordingly
  11. 411 At this point I will insist on replacing the word "ameliorating" with reduction

Adjusted accordingly

  1. 422 remove IFB-088; it is already in the sentence Adjusted accordingly
  2. 442 silencing of GJB1 The gene stated in this line is Gadd34, we change this to italics
  3. 449 overexpressing their genes Adjusted accordingly
  4. 470 mutations cause Cx32 to be retained Adjusted accordingly
  5. 488 abnormally to normally myelinated - the ratio by definition is the quantitative relation between two amounts

The ratio is of abnormally myelinated fibers/total number of fibers, we clarified accordingly

  1. 494 expression of GJB1 or production of Cx32 Adjusted accordingly (we talk about localization of Cx32 in specific subcellular areas)
  2. 503-504 to treat mutant mice producing representative Cx32 mutant proteins, some of which have shown direct interaction with co-produced WT Cx32 Adjusted accordingly
  3. 551 remove replacing Cx32 Adjusted accordingly
  4. 604 What is NFL - this is first time the NFL term is used. It is described in L. 718, but must be explained here. NFL was already first time abbreviated and explained above, in L. 486.
  5. 609 remove for Adjusted accordingly
  6. 661-662 simplify: mitochondrial morphology and mobility Adjusted accordingly
  7. 662 mutations Adjusted accordingly
  8. 664 MFN2 Adjusted accordingly
  9. 692 check grammar Adjusted accordingly
  10. 696 SORD Adjusted accordingly
  11. 702 mutations Adjusted accordingly
  12. 727 HSPA1 and HSPB1 overexpression Adjusted accordingly

L.729 mutated allele expressed Adjusted accordingly

  1. 745 mutations Adjusted accordingly
  2. 861 gene GJB1 Adjusted accordingly

For me in L. 88-90 there is colloquialism, because the genes are the same in all cells of given organism, but they are only differently expressed. I also couldn’t find the term neuro-axonal genes in literature. – please re-write the sentence or convince me, by providing the doi of the publications from the neurological journal(s), that such term is used. Adjusted and clarified accordingly

TABLES

Table 1

  1. Is PSX7 a name for P2X7 encoding gene, if so should be in italic (twice), but I found the name of gene as P2RX7. Adjusted accordingly
  2. deletion of TATA-box of PMP22 gene promotor Adjusted accordingly

Table 2

  1. FIG4 gene in italic Adjusted accordingly
  2. Cx32 production (twice) Adjusted accordingly
  3. SARM1 mutant alleles Please note that there are no mutant SARM1 alleles involved here, only the wild type SARM1 protein which drives axonal degeneration, therefore inhibitors of normal SARM1 protein were used to reduce the NADase activity of normal SARM1 protein.
